# Learning to Count Everything: Transformer-based Trackers are Strong Baselines for Class Agnostic Counting

## Abstract

Class agnostic counting (CAC) is a vision task which can be used to count the total occurrence number of any given reference objects in the query image. The task is usually formulated as density map estimation problem through similarity computation among few image samples of the reference object and the query image. In this paper, we show the the popular and effective similarity computation operation, bilinear similarity, actually share high resemblance with self-attention and cross-attention operations which are widely used in the transformer architecture. Inspired by this observation, since the formulation of visual object tracking task is similar to CAC, we show the advanced attention modules of transformer-based trackers are actually powerful matching tools for the CAC task. These modules allow to learn more distinct features to capture the shared patterns among the query and reference images. In addition, we propose a transformer-based class agnostic counting framework by adapting transformer-based trackers for CAC. We demonstrate the effectiveness of the proposed framework with two state-of-the-art transformer-based trackers, MixFormer and TransT, with extensive experiments and ablation studies. The proposed methods outperform other state-of-the-art methods on the challenging FSC-147 and CARPK datasets and achieve new state-of-the-art performances. The code will be publicly available upon acceptance.

## 1 Introduction

Object counting is a popular research topic in the vision community with a wide spread of applications, including visual surveillance, intelligent agriculture, etc. It aims to count the occurrence number of target objects in an image. The object counting methods can be classified into two major categories: class-specific object counting and class-agnostic counting. For class-specific object counting, it usually focuses on counting a specific category such as car, animal, or people, etc, where crowd counting to count the number of a crowd of people is well studied by Song et al. (2021); Cheng et al. (2022; 2019); Li et al. (2018). However, it requires to train an individual model for each category with tremendous efforts on collecting thousands of training images with annotations and fail to work for unseen classes.

In contrast, class-agnostic counting (CAC) studied by Lu et al. (2018); Ranjan et al. (2021); Yang et al. (2021) arises recently and aims to count any novel objects within the query image, especially for those objects of unseen classes during the training stage. Given several reference object images of the target class, the CAC model is able to predict the number of occurrences within the query image. Current models share a similar network architecture, consisting of a feature extractor, a matching module, and a density head. Once the query and reference feature maps are extracted from the feature extractor, they are fed into the matching module to compute the similarity map followed by a density head to yield a density map estimation. The sum of the values in the density map is used as the final estimated object count. Nevertheless, the major drawbacks of traditional CAC models are their fixed matching framework, which performs template matching in the same fashion regardless of the variations in the patterns of the reference patterns. Moreover, the extracted features for matching are not discriminative enough across categories. Last but not least, traditional models are supervised by pixel-wise root-mean squared error (RMSE) between ground-truth and predicted density map,

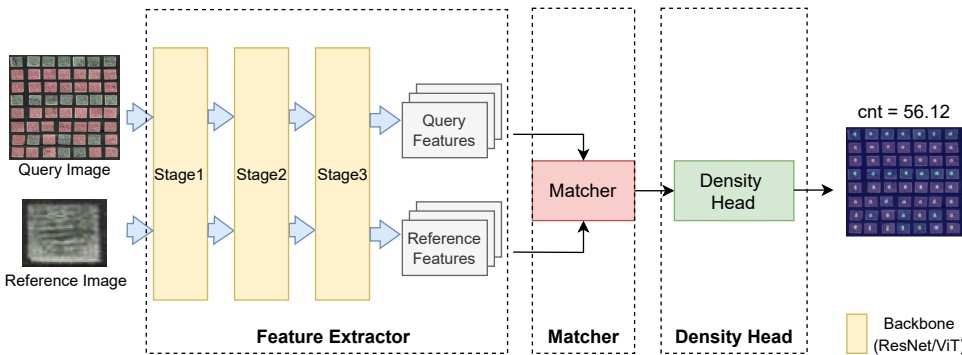

Figure 1: A general pipeline of CAC models consists of a feature extractor, matcher, and density head. Given a query image and at least one reference image, the model learns to predict a density map for counting.

which fails to recognize predictions with respect to each ground-truth object as a unit. The state-of-the-art model BMNet proposed by Shi et al. (2022) alleviates such weaknesses by introducing: (1) bilinear similarity (or dot-product attention) in matching , (2) applying self-attention on feature maps after feature extraction, and (3) additional contrastive loss function. Yet, these revisions are inadequate for more accurate object counting.

In this work, we overcome the above problems of matching and feature extraction by integrating transformer-based object tracking for CAC feature extractor. We first show the connections of the popular bilinear similarity matching module with the prevailing attention modules used by transformer-based methods proposed by Vaswani et al. (2017). In addition, among different transformer-based methods, since the visual object tracking share similar formulation to localize the target object in the query images (or upcoming video frames) as the CAC task, we demonstrate he advanced attention modules of transformer-based trackers are actually powerful matching tools for the CAC task. These attention modules allow to learn more distinct features to capture the shared patterns among the query and reference images. We verify our idea by adapting two state-of-the-art transformer-based trackers, MixFormer by Cui et al. (2022) and TransT by Chen et al. (2021), for the CAC task by replacing the original prediction head for tracking with the U-Net-like density head for density map estimation. With extensive experiments and ablation studies, the results not only demonstrate the effectiveness of the proposed methods but also show the proposed method outperform other state-of-the-art methods on the challenging FSC-147 and CARPK datasets with new state-of-the-art performances. These further show the CAC models based transformer-based trackers are strong baseline for the CAC task.

## 2  RELATED WORKS

Due to a large amount of related works in the literature, we briefly review the relevant ones on class-specific and class-agnostic object counting as follows.

### 2.1  CLASS-SPECIFIC OBJECT COUNTING

Class-specific counting focuses on the task to count the occurrence number of the objects of specific classes, such as crowd counting. In addition, the task is usually approached in two paradigms: detection-based and regression-based methods. The detection-based methods, including the methods proposed by Leibe et al. (2005); Hsieh et al. (2017), perform explicit object detection over the input image using visual object detectors and then get the count. However, similar to the object detection methods, their performances are also sensitive to the situations when the objects are overlapped, occluded, or crowded. To address these problems, the regression methods proposed by Thanasutives et al. (2021); Ma et al. (2021); Cheng et al. (2022) instead predict the density map of input images where each pixel value can be interpreted as the fraction or the confidence level of the target object present in the query image. The sum of these values is then used as the estimated object count. In addition, the ground truth density maps for training are generated by convolving point annotations

of the training images with properly selected Gaussian kernel. Although class-specific counting methods have achieved satisfactory results, they usually require to train an individual network for each class of objects and can only be applied for counting the classes that have been seen in the training data.

## 2.2 CLASS-AGNOSTIC COUNTING

With respect to class-specific object counting, the class-agnostic counting aims to perform object counting for every class. In its setting, the models usually take a query image and several reference object images from the same class as inputs and predict the count of the class appeared in the query image through the density map regression. A typical CAC model consists of three components: a feature extractor, a matcher, and a density head. The model is also supervised by a ground truth density map and mean square loss. Lu et al. (2018) propose GMNet which is the first CAC framework, where the reference and query feature maps are extracted independently through the feature extractor based on the ResNet backbone proposed by He et al. (2016) . These features are then directly exploited as a pixel-wise regression task to perform object counting. To better capture the interactions between the query and the references during the matching stage, Ranjan et al. (2021) propose FamNet which introduces template matching for the CAC task. FamNet enhances the matching framework by convolving the reference feature maps across query feature maps. Each pixel value in the resulting query feature map represents the similarity between the query image and reference images at that specified location and therefore supports better localization. Building upon the existing structures, CFOCNet proposed by Yang et al. (2021) matches query and references feature maps from different stages of backbone for multi-scale strategy. Moreover, CFOCNet integrates self-attention to strengthen the distinct features of the query image. Furthermore, Shi et al. (2022) propose a state-of-the-art CAC model called BMNet which adopts bilinear similarity for matching. The bilinear similarity is a special case of generalized inner product and provide more flexibility for matching than previous methods using convolution. CounTR proposed byLiu et al. (2022) introduces transformer-based architecture into class-agnostic counting. Different from CounTR, we focus on the relation between the transformer tracking and CAC with template matching and propose a general framework for introducing transformer tracker into CAC model.

Similar to CounTR, since high resemblance of the formulation between the bilinear similarity and the attention module of the transformer framework, we believe more advanced attention modules can help the model focus on important patterns shared among the query and reference images and further strengthen the matching for better counting performance. Since the models of object tracking excel at the task of localizing the target objects from the query images which is similar to CAC, the capability of localization have intrigued our interest to integrate with CAC model. Therefore, different from CounTR which uses transformer encoder for CAC, we adapt the transformer-based tackers for CAC since In addition, we select TransT proposed by Chen et al. (2021) and Mixformer proposed by Chen et al. (2021) to illustrate how transformer-based backbones can perform well on CAC model (e.g., BMNet), where both trackers have have been specialized in localizing target objects through strengthening target features through self-attention and cross-attention as means to accurately match and localize target location within the query image.

## 3 METHOD

In this section, we first introduce the general pipeline for class agnostic counting (CAC) and present the connection between transformer-based tracking and CAC. Furthermore, we demonstrate the details of how to adapt transformer-based trackers for the CAC task and to achieve the state-of-the-art performance.

## 3.1 CLASS AGNOSTIC COUNTING

As shown in Figure 1, standard CAC models consist of three components: (1) a feature extractor which is used to extract visual features from the reference and the query images. (2) a cross matching module which performs template matching through computing a similarity map among the references and the query. (3) a density map estimator which estimate the density map based on the similarity map for counting.

Given a query image, $X \in \mathbb{R}^{H_X \times W_X \times 3}$, $K$ reference images, $Z = \{z_i\}_{i=1}^K$, $z_i \in \mathbb{R}^{H_Z \times W_Z \times 3}$, and a ground truth label, $Y \in \mathbb{R}$ or $Y \in \mathbb{R}^{H_X \times W_X}$ (i.e., the number or the density map of the target class.), a CAC model aims to solve the regression problem as follows to predict the number of the references occurring in the query image.

$$\min_{\theta, \phi, \psi} \mathbb{E}_{(X,Z,Y) \sim \mathbb{D}}[\mathcal{L}(R_\theta(S_\phi(F_\psi(X), F_\psi(Z))), Y)], \tag{1}$$

where $F_\psi(\cdot)$ is a pretrained feature extractor (e.g., the backbones of ResNet-50 proposed by He et al. (2016) or vision transformer (ViT) proposed by Dosovitskiy et al. (2021), etc.), $S_\phi(\cdot, \cdot)$ is the matcher to compute the similarity map between the query and the references, $R_\theta(\cdot)$ is the regressor (e.g., fully connected layers) or density map estimator (e.g., transposed convolutions with bilinear upsampling), and $\mathcal{L}(\cdot, \cdot)$ is the loss function (e.g., mean-squared error (MSE)). The predicted density map, $\tilde{Y}$, can be easily converted to the estimated count through computing the expectation, $\mathcal{C}_z = \sum_{i=1}^{H_X} \sum_{j=1}^{W_X} \tilde{Y}(i,j)$, and $\tilde{Y}(i,j)$ represents the fraction of reference objects occurred at the pixel location $(i,j)$.

## 3.2 GENERALIZED MATCHING MODULE

To precisely predict the occurrence count of the reference class in the query image, the prevailing CAC models developed by Yang et al. (2021); Shi et al. (2022); Ranjan et al. (2021) have pointed out the important role of the matcher. Given the query feature $f_q = F_\psi(X) \in \mathbb{R}^{d \times H_q \times W_q}$ and the reference features $f_r = F_\psi(Z) \in \mathbb{R}^{K \times d \times H_r \times W_r}$, prevailing CAC matching frameworks usually realize the template matching as the dot product of the reference features and the query feature to compute the similarity map, $\pi \in \mathbb{R}^{H_q \times W_q}$. Without loss of generality, for the illustration purpose, we follow the setting of BMNet proposed by Shi et al. (2022) in the rest of the paper. Moreover, we reshape $f_q$ as $\tilde{f}_q \in \mathbb{R}^{d \times (H_q \times W_q)}$ and perform the global average pooling upon $f_R$ along the spatial dimension which results in $\tilde{f}_r \in \mathbb{R}^{K \times d}$ and let $\tilde{f}_r(l) \in \mathbb{R}^d$ be the feature of $l$-th reference. Then, we can compute the similarity map, $\pi$ as follows:

$$\pi = \frac{1}{K} \sum_{l=1}^K \tilde{f}_q^T \tilde{f}_r(l), \tag{2}$$

where $\pi$ is further reshape from 1D vector to 2D map for the following density map estimation once the computation is finished. It is worth noting that distinct methods process $\tilde{f}_q$ and $\tilde{f}_r$ differently.

**Bilinear Matching:** The state-of-the-art CAC model, BMNet, further introduces the bilinear matching for better performance which is a special case of generalized inner product.

$$\pi^{BM} = \frac{1}{K} \sum_{l=1}^K (W_q \tilde{f}_q + b_q)^T (W_r \tilde{f}_r(l) + b_r), \tag{3}$$

where $W_q$ and $W_r$ are trainable transformation matrices, and $b_q$ and $b_r$ are trainable biases. By replacing fixed template matching with bilinear matching, it allows to model flexible interactions among the query and reference features. Furthermore, we can also perceive bilinear matching as to project the features onto a latent space and to better captures the similarity among the query and reference features.

**Cross-attention Matching:** Recall the attention module of the transformer which is computed as follows:

$$Attn(\mathcal{Q}, \mathcal{K}, \mathcal{V}) = softmax(\frac{\mathcal{Q}\mathcal{K}^T}{\sqrt{d}})\mathcal{V}, \tag{4}$$

where $\mathcal{Q}$, $\mathcal{K}$, and $\mathcal{V}$ are the query, key, and value matrices, respectively. $d$ is the feature dimension. We can easily find the bilinear similarity shares high resemblance with the dot-product attention computation by letting $\mathcal{Q} = (W_q \tilde{f}_q + b_q)^T$, $\mathcal{K} = (W_r \tilde{f}_r + b_r)^T$, and $\mathcal{V}$ as the identity matrix. Inspired by this observation, we thus propose to replace bilinear matching with self-attention and cross-attention modules in the literature of transformer since this allows us to exploit various advanced attention modules for versatile matching. On the other hand, before computing the final similarity maps, let $\mathcal{Q} = (W_q \tilde{f}_q + b_q)^T$ and both $\mathcal{K}$ and $\mathcal{V}$ as $(W_r \tilde{f}_r + b_r)^T$, and we can stack multiple attention layers to amplify the reference features in regions that share larger similarity between the references and the query for better matching.

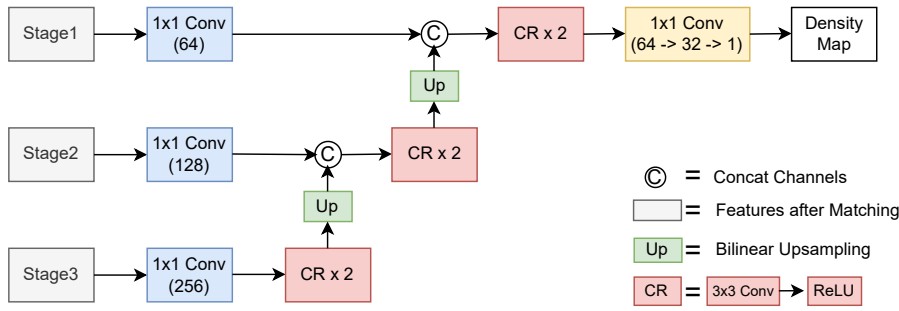

Figure 2: **U-Net-like Fusion of Multi-Scale Features**. We utilize the expansion path (decoder) of U-Net, which fuses features among two adjacent scales with learned convolutions. Numbers within convolution blocks denote the output channels.

### 3.3 TRANSFORMER-BASED TRACKING CAC

As shown in the previous section, attention modules in the transformer frameworks could be used to compute the self-similarity and cross-similarity among the reference features and the query feature. Recently, Liu et al. (2022) propose CounTR to adapt transformer-based architecture onto CAC models and achieve the state-of-the-art performance. We concurrently propose to utilize attention-based modules that enhances both feature extractor and matching. Furthermore, we claim that transformer-based trackers are better candidates for object counting since they not only excel at localizing target objects in the next frame with robust features and matching but also the setting and architecture of transformer-based trackers are similar with existing CAC models, which contain a feature extractor, matcher, and a prediction head. To support our idea, we take two recent state-of-the-art transformer-based trackers, TransT proposed by Chen et al. (2021) and Mixformer proposed by Cui et al. (2022), for further studies to demonstrate the practicality of exploiting the modules of transformer-based trackers for the CAC task. We mainly use their backbones and matchers with the pretrained weights and replace the prediction head for tracking with our density map estimator with the similar design as the decoder of U-Net as shown in Figure 2. More details of the density head can be found in Section 3.4.1. The adapted trackers are denoted as TransTCAC and MixformerCAC. A brief introduction of both trackers is shown in the Appendix.

### 3.4 IMPLEMENTATION DETAILS

Then, we show how to adapt the existing pretrained transformer-based tracker for CAC and the training details to effectively train the models.

#### 3.4.1 MULTI-SCALE ENHANCEMENT FOR MATCHING AND DENSITY MAP ESTIMATION

After features and matching, most of prior works adopt simple density head consisted with several upsampling and convolution operations on single-scale feature maps and achieved acceptable performance. Yet, utilizing feature maps from the last layer/stage as BMNet and stacking convolution layers are insufficient to capture small objects. To preserve the information of fine features from early layers/stages across scales, we respectively perform matching on features across stage 1, 2, and 3, where the feature dimensions are decreased by half as the stage progresses. Then, we utilize U-Net-like expansion path as the density head to fuse query and similarity-related features across scales. After fusing feature maps, we utilize $1 \times 1$ convolutions to compress multi-channel feature maps into a single final density map as shown in Figure 2.

#### 3.4.2 NETWORK ARCHITECTURE

Here, we further explain the details of which network components of TransT and Mixformer are used to substitute for the corresponding ones in the standard CAC models. For TransTCAC, we utilize the ResNet-50 backbone and the featureFusion matcher from TransT. We further tune the featureFusion matcher of TransT by setting the number of the encoder layers as 2 and heads as 1, and compress the input features to the matcher with 256 channels. For MixFormerCAC, we use the pretrained MixFormer-22k and replace the prediction head with our density head for the CAC

task. Regarding the multi-scale version of TransTCAC and MixFormerCAC, we follow the same architecture definition of stage1, 2 and 3 as used in MixFormer proposed by Cui et al. (2022), and for our TransTCAC, we denote the features from layer1, 2 and 3 from the ResNet-50 backbone of TransT as stage1, 2 and 3, respectively.

### 3.4.3 Training Details

**Training Data Preprocessing** FamNet developed by Ranjan et al. (2021) initiated a conventional data preprocessing for the FSC147 dataset. During training stage, reference images are resized to the size of $128 \times 128$. Query images are processed with (1) zero-padding image dimensions to multiples of 32, then (2) zero-pads images to match the dimensions of the largest image within the batch. During evaluation stage, reference images are processed similarly as training but query images are not zero-padded. Given that the image dimensions from the FSC147 dataset vary from 384 to 1,584, excessive paddings are imposed to have the same image sizes within the same batch. The excessive zero paddings could mislead transformer-based models as they learn pixel relationships and localization globally through attention blocks. We then propose a new dataset protocol for training. Instead of adding zero-paddings, query images are (1) randomly flipped horizontally and (2) randomly cropped to the same size of $384 \times 384$. To keep the aspect ratio of the reference images, reference images are then zero-padded to square dimensions and then resized to the resolution of $128 \times 128$. During the evaluation stage, the reference images are processed in a similar fashion as training, while the query images are fed into the model without cropping. Concurrently, CounTR, another transformer-based CAC model, performs the same query image setup by replacing padding with random cropping.

**Training Loss** Currently, most CAC approaches based on density map estimation employ the pixel-wise $L_2$ loss (or MSE loss) to train the models where the ground truth density map is generated by converting the point annotations of each training image through convolving with a Gaussian kernel. However, the counting performance is sensitive to the hyperparameter selection of the Gaussian kernel since it greatly affect the generated ground truth density map. Recently, Ma et al. (2021) propose a generalized loss (GL) which uses optimal transport (OT) to measure the transport cost between the predicted density map and the ground-truth point annotations directly. In addition, it also has shown that the $L_2$ loss usually results in suboptimal solutions and is a special case of GL. Thus, we mainly exploit GL to train the proposed models.

**Optimizer and Training Hyperparameters** Our model is trained with 300 epochs, batch size of 8, optimized with AdamW optimizer, and weight decay of 5e-4. For both TransT and Mixformer, we leverage the tracker's pretrained backbone and matcher onto CAC model. The backbone learning rate is 1e-4 and non-backbone learning rate is 1e-5. We use Pytorch as implementation framework. We as well train on generalized loss function with a ground-truth density map downsampled to $\frac{1}{4}$ ratio in terms of original height and width. We follow the default setting of generalized loss, utilizing mean-absolute error (MAE) as the point-to-point loss and mean-squared error (MSE) for pixel-wise loss. The hyperparameters of the loss function are set as scale = 0.6, reach = 0.5, blur=0.01, $\tau = 0.1$.

## 4 Experiment

In this section, we present the details of experimental setup and then show extensive evaluation results and ablation studies of the proposed approaches on the widely used CAC benchmarks.

### 4.1 Datasets

We utilize two datasets to evaluate our model performance. To compare the model performance, FSC147 dataset collected by Ranjan et al. (2021) is a standard class-agnostic object counting dataset composed of 147 categories and 6,135 images across different scales, each image annotated with three reference objects. Given that the dataset contains decent annotations, we use the FSC147 dataset to train and evaluate our models. To ensure the generalizations of our models on unseen dataset, we also evaluate our FSC147-trained models on CARPK collected by Hsieh et al. (2017), a dataset that contains 1,448 photos of cars in parking lots from a bird-view perspective.

| Methods | Backbone | Pre-trained | Loss | Val MAE | Val RMSE | Test MAE | Test RMSE |
|---|---|---|---|---|---|---|---|
| GMN Lu et al. (2018) | ConvNets | ✗ | MSE | 29.66 | 89.81 | 26.52 | 124.57 |
| FamNet Ranjan et al. (2021) | ConvNets | ✓ | MSE | 23.75 | 69.07 | 22.08 | 99.54 |
| CFOCNet Yang et al. (2021) | ConvNets | ✗ | MSE | 21.19 | 61.41 | 22.10 | 112.71 |
| BMNet Shi et al. (2022) | ConvNets | ✓ | MSE | 19.06 | 67.95 | 16.71 | 103.31 |
| BMNet+ Shi et al. (2022) | ConvNets | ✓ | MSE+CL | 15.74 | 58.53 | 14.62 | 91.83 |
| CounTR Liu et al. (2022) | ViT | ✓ | MSE | 17.40 | 70.33 | 14.12 | 108.01 |
| CounTR+ Liu et al. (2022) | ViT | ✓ | MSE | 13.13 | 49.83 | 11.95 | 91.23 |
| BMNet+* Shi et al. (2022) | ConvNets | ✓ | GL | 15.12 | 53.62 | 13.50 | 91.60 |
| BMNet+* Shi et al. (2022) | ConvNets | ✓ | MSE+CL | 17.37 | 61.36 | 15.75 | 94.44 |
| **MixFormerCAC(Ours)*** | CVT | ✓ | GL | **11.17** | 46.69 | **11.82** | 92.78 |
| **TransTCAC(Ours)*** | ConvNets | ✓ | GL | 11.66 | **43.91** | 12.46 | **82.89** |

Table 1: Comparisons with state-of-the-art CAC models, and CL denotes contrastive loss

## 4.2 EVALUATION METRICS

We follow the same evaluation protocol as the previous CAC works, which evaluate their model performance based on MAE and RMSE. For a dataset of $N$ testing samples, the vector of ground-truth counts is denoted as $C^{gt} \in \mathbb{R}^N$, and the vector of predicted counts denoted as $C^p \in \mathbb{R}^N$.

$$MAE = \frac{1}{N} \sum_{i=1}^{N} |C_i^p - C_i^{gt}|, \ RMSE = \sqrt{\frac{1}{N} \sum_{i=1}^{N} \|C_i^p - C_i^{gt}\|^2} \tag{5}$$

## 4.3 EVALUATION RESULTS WITH OTHER STATE-OF-THE-ART CAC MODELS

**Model Comparison** We compare our models with other CAC models on the FSC147 validation and test sets. For the 3-shot counting task, we compare the proposed models TransTCAC and Mix-formerCAC with recent CAC models. We are specifically interested in the comparison with BMNet+ (officially current state-of-the-art) and CounTR (recently released state-of-the-art). Given that our model does not perform test-time normalization, we categorize CounTR into two settings, where the plus version integrates test-time normalization. We denote * beside the method name as the model performance under our dataset protocol. We especially run BMNet on our dataset protocol for the fair comparison purpose.

**Quantitative Results** As shown in Table 1, the proposed CAC models based on transformer-based trackers achieve the state-of-the-art performance comparing to BMNet, BMNet+ and CounTR. In comparison to BMNet+, the official state-of-the-art, our models yield an improvement of $29.03\%$ on VAL MAE, $24.98\%$ on VAL RMSE, $19.15\%$ on TEST MAE, and $9.74\%$ on TEST RMSE. Even without multi-scale setting, our models are able to achieve the best performance, proving the validity of exploiting transformer-based trackers as baselines of CAC task. Without adopting the same test-time normalization strategy as CounTR+, the proposed approach models still outperform CounTR and CounTR+.

**Qualitative Visualizations** As shown in Figure 3, our models are able to locate objects more accurately with higher confidence, especially in dense scenes. While BMNet+ is able to identify position of target objects, the confidence regions are spread more widely. On the other hand, our models are capable of placing point annotations right at the center of the target objects with smaller radius, modelling with less ambiguity. When inspecting the first row in Figure 3, the proposed models are better at locating birds in dense scenes and predict a density map that aligns closer to the ground truth and wouldn't create incorrectly distributed patches.

**Cross-Dataset Generalization** A strong CAC model should excel in counting objects on scenarios that differ from the training dataset in terms of perspectives, scales, illumination, etc. It should be able to perform well on unseen scenarios without additional exposure on similar scenarios. To evaluate the ability to generalize on other datasets, we train the proposed method on the FSC147 dataset and then evaluate on CARPK testing set without the use of fine-tuning on the CARPK training set. Specifically, since the CARPK dataset mainly consists of car images, we remove the car category from the FSC147 dataset during training stage. As shown in Table 4.3, both of our models outperform the non-fine-tuned baseline. Moreover, MixFormerCAC is able to win over other fine-tuned

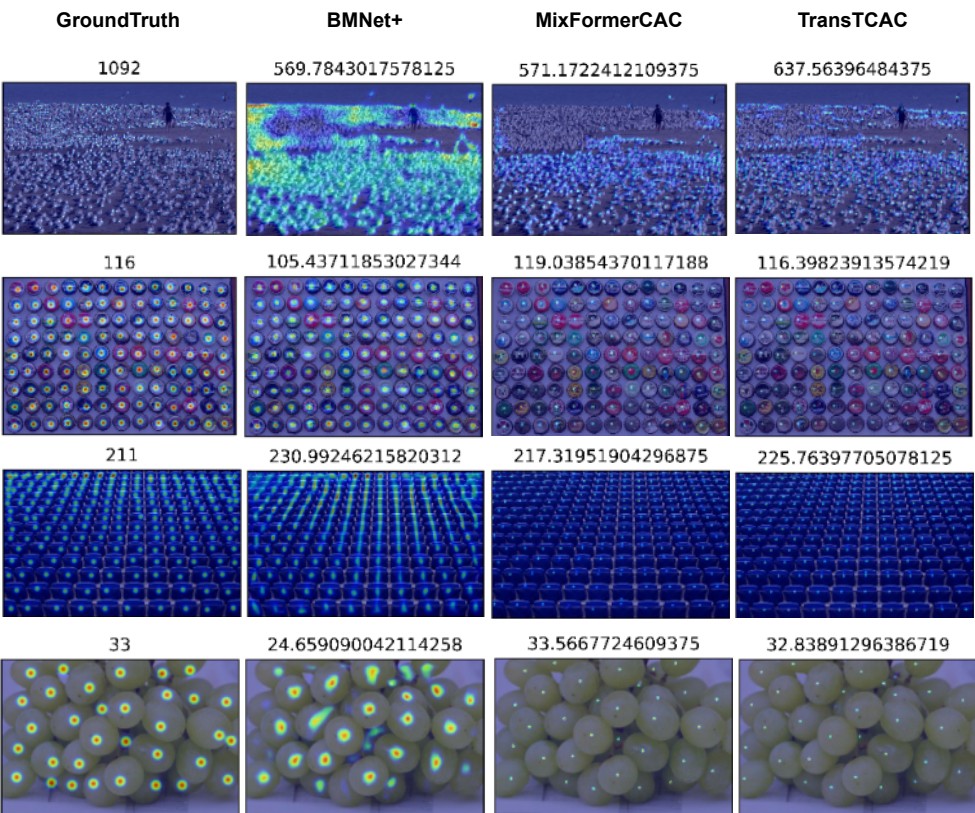

| GroundTruth | BMNet+ | MixFormerCAC | TransTCAC |

Figure 3: **Qualitative Visualization**. We visualize the predicted density map results of the proposed approaches with BMNet+ on the FSC147 testing dataset. The leftmost column shows the ground-truth density maps overlaid onto the original images.

models and achieve the state-of-the-art performance. This further demonstrate the proposed models better excel in various counting scenarios than other methods.

| Model | Fine-tuned | MAE | RMSE |
|---|---|---|---|
| BMNet Shi et al. (2022) | ✓ | 8.05 | 9.7 |
| BMNet+ Shi et al. (2022) | ✓ | 5.76 | 7.83 |
| CounTR Liu et al. (2022) | ✓ | 5.75 | 7.45 |
| BMNet Shi et al. (2022) | × | 14.61 | 24.60 |
| BMNet+ Shi et al. (2022) | × | 10.44 | 13.77 |
| **TransTCAC(Ours)** | × | 9.61 | 12.65 |
| **MixFormerCAC(Ours)** | × | **3.6** | **4.67** |

Table 2: Cross-dataset evaluation results compared with other state-of-the-art models where 'fine-tuned' means the model is further fine-tuned using the CARPK training data.

## 4.4 IMPACT OF NUMBER OF REFERENCES IMAGES

We investigate our model performance in relation to the number of reference images, where $n \leq 3$. As shown in Table 3, the results indicate that one reference image is adequate for the proposed models to reach comparable performance with current state-of-the-art performance, given the ability of the attention-based modules within transformer-based tracker to extract representative reference features through global interactions among query and reference features. Moreover, the multi-scale fusion also allows the model to effectively capture objects of different sizes. Given more reference images, the proposed models are able to yield better results and excel in each of scenarios of $n$ reference images.

| n | MixFormerCAC(Ours) | | TransTCAC(Ours) | | BMNet+ Shi et al. (2022) | |
|---|---|---|---|---|---|---|
| | Val MAE | Val RMSE | Val MAE | Val RMSE | Val MAE | Val RMSE |
| 1 | 14.77 | 62.45 | 12.82 | 45.14 | 17.89 | 61.12 |
| 2 | 12.65 | 52.68 | 12.17 | 45.39 | 16.03 | 58.65 |
| 3 | **11.17** | **46.69** | **11.66** | **43.91** | 15.74 | 58.53 |
| n | MixFormerCAC(Ours) | | TransTCAC(Ours) | | BMNet+ Shi et al. (2022) | |
| | Test MAE | Test RMSE | Test MAE | Test RMSE | Test MAE | Test RMSE |
| 1 | 12.63 | 106.41 | 13.62 | 82.44 | 16.89 | 96.65 |
| 2 | 11.90 | 97.21 | 12.75 | 85.37 | 16.16 | 97.18 |
| 3 | **11.82** | **92.78** | **12.46** | **82.89** | 14.62 | 91.83 |

Table 3: Evaluation results of the proposed methods using different numbers of reference images. Bold font indicates the best model performance of MixFormerCAC and TransTCAC.

| No. | Feature | Val MAE | Val RMSE | Test MAE | Test RMSE |
|---|---|---|---|---|---|
| S1 | B | 18.79 | 66.9 | 15.82 | 104.6 |
| S2 | B+GL | 12.29 | 56.88 | 13.09 | 99.58 |
| S3 | B+GL+MS | **11.66** | **43.91** | **12.46** | **82.89** |

Table 4: **Ablation studies** It shows the ablation results of the proposed TransTCAC where 'B' denotes using backbone and matcher of TransT followed by a simple density head using the features from the last layer, 'GL' denotes replacing the MSE loss with the generalized loss, and 'MS' means to use multi-scale features instead with the U-Net-like fusion.

## 4.5 ABLATION STUDIES

As the architecture of TransTCAC is similar to BMNet, we perform ablation studies on TransTCAC to examine the functionality of our proposed framework in relation to the baseline model BMNet. (1) **Transformer-based backbone**: We integrate transformer-based backbone (ResNet-50) and matcher with the CAC density head. The essential difference between S1 and BMNet is the substitution of the BMNet matcher with the TransT matcher. The results show that the performance of TransTCAC in S1 surpasses BMNet by 0.89 in test MAE and 0.29 in val MAE, suggesting the strength of transformer-tracker matching. (2) **Generalized Loss**: The introduction of generalized loss yields a relative improvement of 6.5 in val MAE and 2.73 in test MAE. The loss function provides stronger supervision, therefore allowing TransTCAC to effectively exploit attention modules and identify regions of interests. (3) **Multi-scale Feature with U-Net-like Fusion**: The addition of multi-scale feature extraction with U-Net-like fusion allow to capture the objects across scales. It thus enables TransT to further improve the counting performance by a large margin and to achieve 12.91 in val RMSE and 16.69 in test RMSE.

## 5 CONCLUSION

In this work, we show the connection of the popular and effective bilinear similarity matching for the CAC task with the attention modules which are widely used in the transformer-based methods. Furthermore, inspired by this finding, since the visual object tracking shares similar components with CAC, we demonstrate the advanced self- and cross-attention modules used in transformer-based trackers are powerful matching modules for the CAC tasks and can greatly help the model learn more distinct features to capture the shared pattern among the query and reference images. We demonstrate the effectiveness of this idea by adapting two state-of-the-art transformer-based trackers and achieve new state-of-the-art performances for the CAC task. Moreover, we also find that the widely used zero-padding strategy for the query images during training as used in previous CAC models hurts the inference performance when we adapt the transformer-based trackers for CAC. We instead propose a random cropping strategy for the query images to avoid performing any zero-padding for improved performance. With extensive experiments and ablation studies, we show that the CAC models based on transformer-based tracker are strong baselines for the CAC task.

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
