# OpenReview forum: "Learning to Count Everything: Transformer-based Trackers are Strong Baselines for Class Agnostic Counting"
_ICLR.cc/2023/Conference — Submitted to ICLR 2023_

### Official Review · Reviewer_zJ5d · 2022-10-23

**Confidence:** 3
**Correctness:** 2
**Technical Novelty And Significance:** 3
**Empirical Novelty And Significance:** 3
**Recommendation:** 5

**Clarity, Quality, Novelty And Reproducibility:**

Overall, the paper is clear and provide quality content. The methodology appear relatively novel, but the contributions are not clearly stated. The authors promised reproducible code upon publication.

The paper missed a proof-reading. Some sentences are incomplete or missing “...we adapt the transformer-based trackers for CAC since In addition, we select TransT …” (related work), as well as some prepositions all across the paper.

The exact contributions of this paper are not clearly stated, and most contributions appear to be borrowed from previous works in different fields. The paper would benefit from a clear statement of the contribution.

The hyperparameters scale, reach, blur and \tau are not defined in the paper.

The dataset FSC147 is named differently in the paper (FSC-147 vs FSC147).

The aggregation of the results obtained from multiple references (e.g. for the 3-shot counting task) are not explained in the paper, hence is it unclear how the results from the multiple references are merged together.

Figure 3: Would it be possible to visualize the qualitative results for the ablation study? It would validate the sharpness of the density map introduced by the GL based on OT.

Figure 3: The significant figure should be truncated or averaged.

Table 2: What are the performances on CARPK once fine-tuned? Also, the table is missing the SOTA method “An Accurate Car Counting in Aerial Images Based on Convolutional Neural Networks”, Kilic et al., (2021) that appears SOTA on CARPK.



**Strength And Weaknesses:**

Strengths:

The method presented by the authors reached impressive state-of-the-art performances on the FCS-147 dataset for CAC.
The method is able to transfer to CARPK (with state-of-the-art performances) without the need for any extra fine-tuning.
The similarity of the architectures for CAC with the architectures for VOT is interesting and worth noting.

Weaknesses:

The similarities drawn between the CAC and VOT tasks omitted an important analysis. In particular, VOT methods are trained to discriminate similar objects (same class) from the same image, while CAC actually wants to look at those exact objects. In VOT, those extra samples in a frame are referred to background clutter, and previous works focus on learning to discard those distractors (e.g. “Context-aware correlation filter tracking”, Mueller et al., CVPR’17 / “Distractor-aware siamese networks for visual object tracking”, Zhu et al., CVPR’18 / “Exploring Motion Information for Distractor Suppression in Visual Tracking”, Liu et al., CVPR’22). Because of the nature of the tasks, I would respectfully disagree with the authors who state that “the formulation of visual object tracking task is similar to CAC” (abstract), “visual object tracking formulation share similar formulation [...] as the CAC task” (introduction), “object tracking excel at the task of localizing the target objects from the query image which is similar to CAC” (related work), and “the setting and architecture of transformer-based trackers are similar with existing CAC models” (methodology) among other places in the remainder of the paper.


While the CAC and VOT architectures show significant similarities, the knowledge a tracker is expected to learn is different from the knowledge needed for a CAC task. The learning process aims at different goals, and as a result, one would not expect the knowledge pre-trained in a VOT task to transfer to a CAC task. TransT and MixFormer were both pre-trained on a tracking task, and the paper misses an analysis comparing the performance with and without transferring the negative knowledge learned from the tracking task.


From the ablation study, it appears that most improvement originates from the optimal transport technique borrowed from Ma et al., and very little from the attention mechanisms, and tracker architecture. As a result, I would recommend putting the focus on that generalized loss, rather than biasing the reader into believing that the improvement originates from expensive transformer-based mechanisms or a tracking formulation. Moreover, the paper only mentions that loss without properly defining it. It is a pity and frustrating for the reader to not fully grasp and understand that loss, that contributes to most of the improvement in the ablation study.


**Summary Of The Paper:**

The authors of this paper present a novel perspective for the task of Class Agnostic Counting (CAC), drawing similarities with the task of Visual Object Tracking (VOT), where a pattern (VOT: patch / CAC: reference) is localized in an image (VOT: search space / CAC: query image).

In particular, the authors leverage:
 - previous finding in VOT models including self- and cross- attention mechanisms (from TransT, Chen et al. (2021) and MixFormer, Cui et al. (2022)),
 - previous findings in CAC models, including the general structure of the models and a Bilinear Matching (from BMNet, Shi et al. (2022)),
 - previous finding in crowd counting models, including the generalized loss based on optimal transport (from SDNet, Ma et al. (2021)), and
 - novel modules such as the Multi-Scale enhancement for matching and density map estimation
 - novel preprocessing of the dataset (inspired by FamNet, Ranjan et al. (2021))

to build a CAC model showing state-of-the-art performances on the datasets/benchmarks FSC-147 and CARPK.

The authors provide a complete analysis of their method, with an ablation study for all the modules they have integrated.


**Summary Of The Review:**

The analysis of the similarities between tracking architectures and CAC architectures is interesting to be noted and analyzed. It appears that the methodology leads to SOTA results on the latest benchmarks, which is remarkable.

Yet, the authors claim that transformer-based trackers are strong baselines for CAC, but it appears that most of the improvement in the experiments originate from the Generalized Loss based on Optimal Transport, introduced by Ma et al. (2021) in a different task. The ablation study confirms such a statement. It is misleading to claim that the results originate from self-/cross- attention mechanisms and the transferability of trackers’ knowledge to the task of class-agnostic counting.

The paper is very borderline, as I believe the analysis with trackers is a fresh perspective, yet it needs further investigations as it is misleading in the current form. As a result, I would recommend a score of 5.

---

> ### Author Response · Authors · 2022-11-19
> **Reply to the comments of the reviewer**
>
> We sincerely thanks the reviewer for your detailed and constructive feedback. We will adjust our title and related wordings in the main paper as suggested to avoid misunderstanding and confusion. Although the proposed method adapts the transformer-based tracker for the class agnostic counting (CAC), we train the model from the ImageNet pretrained model as other compared state-of-the-art baseline methods for the CAC task without pretraining upon any visual object tracking datasets. We will make it more clear in the final paper.
>
> The generalized loss (GL) does provide significantly better supervisory signal for transformer-tracker-based backbone for class-agnostic counting than the plain MSE loss. However, we also show the results of BMNet+ with GL which does not result in significant performance boost as the performances of BMNet+ with MSE and Contrastive losses. Thus, the proper combination of the architecture and the loss also play an important role for the final performance. We will make it clear as suggested. Thanks for your profound feedbacks.
>
> We will make corresponding revision for the manuscript as suggested. Very thanks the reviewer for your detailed proofreading.
>
> We describe the details of $k$ reference shots as follows:
> Given $k$ references and the query images, the dimension of reference and query features are $(k, H_r, W_r, 3)$ and $(k, H_q, W_q, 3)$.
>
> For TransTCAC, the reference and query feature maps dimension before the matcher are $ \mathbb{R}^{k \times H^s_r \times W^s_r \times C} $ and
> $ \mathbb{R}^{H^s_q \times W^s_q \times C} $ respectively, where $s$ is the stage of the backbone and $C$ is the hidden dimension. We then flatten the references feature maps to $ f_r(s) \in \mathbb{R}^{(k \times H^s_r \times W^s_r) \times C }$, and query feature maps to $ f_q(s) \in \mathbb{R}^{(H^s_q \times W^s_q) \times C }$.
>
> Self-attention(ECA) is first performed on query and reference features, yielding the same dimension. For the cross-attention mechanism(CFA), we'll demonstrate the incorporation of $k$ reference features through the top cross-feature augment(CFA) module. It receives $ \mathcal{Q} = f_r \in \mathbb{R}^{(k \times H^s_r \times W^s_r) \times C}$, $\mathcal{K} = f_q \in \mathbb{R}^{(H^s_q \times W^s_q) \times C } $, $\mathcal{V} = f_q \in \mathbb{R}^{(H^s_q \times  W^s_q) \times C }$.
>
> \begin{equation}Attn(\mathcal{Q}, \mathcal{K}, \mathcal{V}) = softmax(\frac{\mathcal{Q}\mathcal{K}^T}{\sqrt{C}})\mathcal{V} = \mathcal{W}\mathcal{V},\end{equation}
> where $\mathcal{W}=softmax(\frac{\mathcal{Q}\mathcal{K}^T}{\sqrt{C}})$ and therefore shares the same dimension as
> $ \mathcal{Q}\mathcal{K}^T \in \mathbb{R}^{(k \times H^s_r \times W^s_r) \times (H^s_q \times W^s_q)} $. $Attn(\mathcal{Q}, \mathcal{K}, \mathcal{V}) = \mathcal{W}\mathcal{V} \in \mathbb{R}^{(k \times H^s_r \times W^s_r) \times C}$ which matches the dimension of $ f_r $. Within cross-attention, the first dimension of $ \mathcal{K}, \mathcal{V} $ will be merged and omitted during multiplication. Similarly, when applying the bottom CFA module with $\mathcal{Q} = f_q, \mathcal{Q} = f_r, \mathcal{V} = f_r$,  any arbritrary $ k $ values of $ k $ reference images can be incorporated into matching.
>
> For MixFormerCAC in each stage, we perform the convolution embedding to the references and the query, the embedding tokens are flatten and concatenated with the size of $(k \times H^s_r \times W^s_r + H^s_q \times W^s_q,C)$. The concatenated tokens pass through multiple target-search Mixed-Attention Module(MAM) to perform both extraction and matching mechanism. The MAM simultaneously performs self-attention on the reference features on cross-attention on query features. The self-attention intuitively preserves the reference feature dimensions. For cross attention, it receives concatenated tokens $f_{cat} \in \mathbb{R}^{(k \times H^s_r \times W^s_r + H^s_q \times W^s_q) \times C }$ as $\mathcal{K}, \mathcal{V}$ field in attention. Given $\mathcal{K} = f_{cat}$, $\mathcal{V} = f_{cat}$, $\mathcal{Q} = f_q$, the attention weight $\mathcal{Q}\mathcal{K}^T \in \mathbb{R}^{(H^s_q \times W^s_q) \times (k \times H^s_r \times W^s_r + H^s_q \times W^s_q)}$. This results in the output feature of cross attention to be $ \mathbb{R}^{(H^s_q \times W^s_q) \times C}$, the same dimension as the query feature before attention mechanism.

---

> > ### Author Response · Authors · 2022-11-19
> > **Reply to the comments of the reviewer**
> >
> > We further show some qualitative results of BMNet+ (MSE+CL) and BMNet+(GL) with the proposed two methods as the following url.
> > https://github.com/rt6prgcr/Transformer-Tracker-CAC/blob/main/cac_comparison.pdf
> >
> > From the Figure, we can see directly employing the generalized loss can not always yield better predictions. Thus, this further demonstrates the combination of both architecture and loss function is important as described above.
> >
> > We have not done the results of CARPK with finetuning. We will include the results as compared with Kilic et al., (2021) in the final paper. Very thanks the reviewer for the reference suggestions!
> >
> > ----------
> > To prove the strength of the transformer-tracker-based attention modules on class-agnostic object counting, we visualize the attention weights of final cross-attention module in each stage of our multi-scale framework.
> > For TransTCAC, we'll extract the weights of decoder CFA's cross-attention module.
> > After performing the attention operation, we get the attention map of $(L_q,L_r)$, which $L_q=  H^q_r \times W^q_r$ and $L_r =  k \times H^s_r \times W^s_r$ are the flattened length of query and reference in each stage. We choose the center of reference feature maps as reference points to visualze the attention map, that is, the selected attention map is the size of $(L_q,1)$, we reshape it to the $(H^s_q,W^s_q)$, which represents the size of input feature map in $s$ stage.
> > For Stage 1, the attention weights show the model focuses more on the contour of the query. For Stage 2 and 3, the models can locate the target objects.
> >
> > The resulting figure can be found at
> > | Method | URL|
> > |------|-----|
> > |TransTCAC: |https://github.com/rt6prgcr/Transformer-Tracker-CAC/blob/main/transt_attn_vis.pdf|
> > |MixformerCAC: |https://github.com/rt6prgcr/Transformer-Tracker-CAC/blob/main/mix_attn_vis.pdf|
> > From the figures, we can see the advanced attention module can effectively help capture the target objects during different stages.

---

### Official Review · Reviewer_eXTk · 2022-10-24

**Confidence:** 4
**Clarity, Quality, Novelty And Reproducibility:** The authors promise to release the co…
**Correctness:** 4
**Technical Novelty And Significance:** 2
**Empirical Novelty And Significance:** 2
**Recommendation:** 3

**Strength And Weaknesses:**


Weaknesses

-The largest concern is the limited contribution. The authors directly apply two existing trackers to fix the counting task with minor changes by extending them to multi-scale architecture with an off-the-shelf generalized loss.  Besides, the core idea of transferring the CAC problem to a matching task between reference and query is the same with pervious arts. This paper only replaces the simple inner-production with more complex Transformer modules. I can't see any new insights for solving counting tasks in this paper.

-Based on the comparison between Table 1 and Table 4, the improvement largely benefits from the used GL loss. The baseline model by using only TransTCAC is even not comparable with 'BMNet+* Shi et al. (2022)'. The current results can't support the advantage by using Transformer Trackers.

-The experiments are not fair according to the current statement since the input resolution, model size, computational cost and the usage of multi-scale structure are missing in Table 1.

-The statement 'Even without multi-scale setting, our models are able to achieve the best performance, proving the validity of exploiting transformer-based trackers as baselines of CAC task' in Quantitative Results of Sec. 4.3  is confusing. It seems that the listed TransTCAC(Ours)* already used the multi-scale setting according to Table 4.



**Summary Of The Paper:**

This paper proposed to apply two existing Transformer-based trackers to class agnostic counting and evaluated their performance on public datasets.

**Summary Of The Review:**

This paper's novelty by applying siamese trackers is incremental. The used networks are directly borrowed from existing tracking methods without adaptively redesign for counting task. Besides, the performance improvement largely benefit from Transformer-based attention block which brings more computational cost. Based on these concerns, I tend to reject this paper.

---

> ### Author Response · Authors · 2022-11-19
> **Reply to review comments**
>
> We sincerely thanks for the profound feedbacks from the reviewer. We would like to elaborate our main goal is to show that the combination of both advanced self-attention and cross-attention with proper loss functions for supervision could further benefit the matching of the reference objects and the search image even better than plain ViT-based method, such as CounTR. We will make it clear in the final paper.
>
> We thanks the reviewer for the feedback of the comparison with BMNet+ where BMNet+ utilizes other techniques, such as contrastive loss and scale embeddings as compared to BMNet and the proposed method. We compare the proposed models with BMNet given that we didn't employ the additional techniques,  If we compare TransTCAC's data (row S1, table 4) with BMNet's (row 4, table 1), TransCAC's performance results in better performance, which suggests the validity of transformer-tracker-based backbones. Moreover, if we look upon the results of CounTR (row 6, table 1), the performance of the proposed method without employing test-time normalization is even better than or comparable to CounTR with additional test-time normalization techniques to boost their performance.
>
> If we applied similar test-time normalization technique as CounTR, the performances of the proposed method can be further improved.
>
> With our custom test-time normalization which is similar as CounTR, the performances of the proposed method outperform other SOTA baselines, including BMNet+.
>  | Method |  Val MAE | Val RMSE | Test MAE | Test RMSE|
>  | ---------  |  --------  |   ---------  |    ---------  |    ---------  |
>  | MixFormerCAC| 9.73 | 44.45 | 12.1 | 87.29|
> | TransTCAC| 11.46 | 40.66 | 13.23 | 63.95|
>
> Thanks for the reviewer of the feedback for the setting for comparisons in Table 1. The descriptions of the input resolution and the usage of multi-scale structure in Table 1 can be found in the 3.4.1(Multi-Scale enhancement), 3.4.2(Network Architecture), and 3.4.3(Training Details - Training Data-Preprocessing). We will make it more clear in the final paper. We further show the model size and computational costs as follows:
>
> |                Method            |      #Params    | Inference Time |
> |    ----------------------------- |  -----------------  |-------------------- |
> |  MixFormerCAC (Ours)  |  32,611,625     |       53ms        |
> |  TransTCAC (Ours)        |  19,535,809     |    227ms         |
> |  CounTR                        |   98,953,345    |    100ms         |
> |  BMNet+                         |   12,862,218   |        27ms        |
>
> For the issue of without using multi-scale setting, we show the performance of TransTCAC with the single scale setting on the ablation study, which is in Table 4 Row S2. As compared with other methods in Table 1, we achieve the best performances in terms of validation and test MAE without the assistance of multi-scale setting. We will make it clear in the final paper. Very thanks for the feedback.
>
> ----------
> To prove the strength of the transformer-tracker-based attention modules on class-agnostic object counting, we visualize the attention weights of final cross-attention module in each stage of our multi-scale framework.
> For TransTCAC, we'll extract the weights of decoder CFA's cross-attention module.
> After performing the attention operation, we get the attention map of ($L_q$,$L_r$), which $L_q=  H^q_r \times W^q_r$ and $L_r =  k \times H^s_r \times W^s_r$ are the flattened length of query and reference in each stage. We choose the center of reference feature maps as reference points to visualze the attention map, that is, the selected attention map is the size of $(L_q,1)$, we reshape it to the $(H^s_q,W^s_q)$, which represents the size of input feature map in $s$ stage.
> For Stage 1, the attention weights shows the model focuses more on the contour of the query. For Stage 2 and 3, the models can locate the target objects.
>
> The resulting figure can be found at
> | Method | URL|
> |------|-----|
> |TransTCAC: |https://github.com/rt6prgcr/Transformer-Tracker-CAC/blob/main/transt_attn_vis.pdf|
> |MixformerCAC: |https://github.com/rt6prgcr/Transformer-Tracker-CAC/blob/main/mix_attn_vis.pdf|
> From the figures, we can see the advanced attention module can effectively help capture the target objects during different stages.

---

### Official Review · Reviewer_8qRK · 2022-10-30

**Confidence:** 5
**Correctness:** 4
**Technical Novelty And Significance:** 2
**Empirical Novelty And Significance:** 2
**Recommendation:** 5

**Clarity, Quality, Novelty And Reproducibility:**

Paper is clearly written, and the proposed approach is well explained. Paper has somewhat limited novelty, given paper follows previous works like CounTR and BMNet.

**Strength And Weaknesses:**

Strengths:
The main claim of the paper, that Transformer based approaches designed for object tracking are suitable for exemplar based CAC is reasonable. And the significant boost in performance achieved by the proposed approach provides some weight to the claim.

Weaknesses:
1. Original technical contribution of the paper is somewhat limited, since previous work like CounTR  has already advocated the use of Transformer for Class Agnostic Counting task.
2. Most of the previous approaches on FSC-147 use Imagenet pretrained backbone such as Resnet-50 and ViT. Proposed approach uses backbones trained on Visual Object tracking datasets. This extra training data being used could be one of the reasons behind the superior performance of the proposed approach.
3. Ablation study in table 4 would have been more complete had there been a row corresponding to ViT backbone (similar to the first row S1, but using ViT backbone instead of TransTCAC backbone).


**Summary Of The Paper:**

Paper tackles the task of Class Agnostic Counting (CAC) in a few-shot setting, which involves predicting the overall count for the object of interest in an image, given few exemplars of object of interest from the same image. Previous works on CAC such as Ranjan et al, FamNet, rely on correlation between the exemplar features and input image features to predict the density map (heatmap), and the sum of the density map serves as the overall count. Similar to the work of Shi el al, CountTR, authors replace the Resnet-50 backbone from earlier works with Transformer based backbone. Unlike CountTR, which uses Imagenet pretrained ViT backbone, authors use Transformer backbones designed for Visual Object tracking task. Drawing inspiration from BMNet, which uses bilinear matching between the exemplar features and image features, authors first draw similarities between bilinear matching and attention, and propose a attention based approach for matching the two sets of features.

**Summary Of The Review:**

My main concern with the paper is the limited novelty of the proposed approach, as stated in the Strengths and Weaknesses section.

---

> ### Author Response · Authors · 2022-11-18
> **Reply to review comments**
>
> We sincerely thanks for the profound feedbacks from the reviewer. We would like to first elaborate the issue for the
> second weakness. Although the proposed method adapts the transformer-based tracker for the class agnostic counting (CAC),
> we train the model from the ImageNet pretrained model as other compared state-of-the-art baseline methods for the CAC task
> without pretraining upon any visual object tracking datasets. We will make it more clear in the final paper.
>
> CounTR is based on ViT architecture. We would like to highlight the difference between our approach and counTR as follows:
> 1.  CounTR use two-stage training, which costs more time. The training consists of 300 epochs of pre-training and 1000 epochs fine-tuning on FSC147. In contrast, our model is only trained on FSC147 for 300 epochs without pre-training.
> 2.  The number of CounTR's  parameters is more than the proposed approach where the numbers of parameters of our two proposed models, TransTCAC and MixformerCAC are less than $\frac{1}{3}$ of CounTR.
> 3. The performances of the proposed methods without employing test-time normalization are better than or comparable to CounTR with employing test-time normalization which can greatly improve the performance when the sizes of objects are small with respect to the training dataset as CounTR.
>
> For the FSC-147 dataset, we show the performances of the proposed approach without test-time normalization and CounTR with test-time normalization as follows:
> |     Method                     | Backbone | Val MAE| Val RMSE | Test  MAE | Test RMSE |   #params   |
> |------------------------------  | ------------- | ----------- |  -------------- | ------------- | --------------- | ----------------|
> |CounTR                        |      ViT      |  13.13    |     49.83      |     11.95    |      91.23     |   98,953,345 |
> |MixFormerCAC (Ours)  |    CVT      |  11.17    |     46.69      |     11.82    |      92.78     |   32,611,625 |
> |TransTCAC  (Ours)       | ConvNets |  11.66   |     43.91       |     12.46   |      82.89     |   19,535,809  |

---

### Decision · Program_Chairs · 2023-01-20

**Decision:**

Reject

**Justification For Why Not Higher Score:**

The contribution of the article is very limited.

**Justification For Why Not Lower Score:**

N/A

**Metareview: Summary, Strengths And Weaknesses:**

The article proposes a counting method agnostic of the class. It is based on transformers and achieves state-of-the-art in several datasets. The method also shows transfer capabilities to other datasets.

The main problem with the method is that the contribution is not clear. Other articles, like CounTR, have already proposed using transformers for this problem. It seems that the improvements come from pretraining on more datasets. Moreover, there are other problems, such as more ablation results needed, difficulties in comparison with baseline models, not fair experiments, confusing statements, and missing analysis.

The authors have addressed some of the reviewer's concerns. However, the contribution of the article is limited, and accordingly, I suggest rejecting this article.